# Revealing the Bioactivities of *Physalia physalis* Venom Using *Drosophila* as a Model

**DOI:** 10.3390/toxins16110491

**Published:** 2024-11-15

**Authors:** Zuzanna Tomkielska, Jorge Frias, Nelson Simões, Bernardo P. de Bastos, Javier Fidalgo, Ana Casas, Hugo Almeida, Duarte Toubarro

**Affiliations:** 1Center of Biotechnology of Azores (CBA), University of the Azores, 9500-321 Ponta Delgada, Portugal; zuziatomkielska@gmail.com (Z.T.); jorge.mv.frias@uac.pt (J.F.); nelson.jo.simoes@uac.pt (N.S.); 2Mesosystem Investigação & Investimentos by Spinpark, Barco, 4805-017 Guimarães, Portugal; bernardopbastos@outlook.pt (B.P.d.B.); id@mesosystem.com (J.F.); ana@mesosystem.com (A.C.); hperas5@hotmail.com (H.A.); 3Department of Hospitality and Tourism, Faculty of Social Sciences and Technology, Universidade Europeia, 1500-210 Lisbon, Portugal; 4CETRAD-UE—Center for Transdisciplinary Development Studies, Research Group for Tourism & Sustainability, Quinta de Prados, Pole II, ECHS, Room 1.14, 5000-801 Vila Real, Portugal; 5Associate Laboratory i4HB Institute for Health and Bioeconomy, Faculty of Pharmacy, University of Porto, 4050-313 Porto, Portugal; 6UCIBIO Laboratory of Pharmaceutical Technology, Faculty of Pharmacy, University of Porto, 4050-313 Porto, Portugal

**Keywords:** *Physalia physalis*, venom, *Drosophila melanogaster*, neurotoxicity, nociception, heat avoidance, hyperexcitability, paralysis

## Abstract

*Physalia physalis*, commonly known as the Portuguese Man o’ War, is one of the most venomous members of the Cnidaria yet is poorly understood. This article investigates the toxicity of *P. physalis* venom by assessing its behavioral and toxicological effects on *Drosophila melanogaster*. The venom administered orally revealed dose- and time-dependent mortality, with an LD50 of 67.4 μg per fly. At sublethal doses, the treated flies displayed uncoordinated movement and fell when attempting to climb. Real-time analysis of flies exposed to the venom revealed hyperexcitability followed by paralysis, with phenotypes similar to those observed in vertebrate models. The venom was shown to be non-thermolabile, as no significant differences in behavior and locomotion were observed between flies exposed to untreated or thermally treated venom. The circadian rhythm alterations, the enhanced light attraction, and the reduction in heat avoidance suggest altered neuronal function. This abnormal behavior indicates that the venom contains bioactive molecules, opening avenues for discovering new compounds with potential for pharmacological applications.

## 1. Introduction

*Physalia physalis*, commonly known as the Portuguese Man o’ War, is one of the most remarkable yet poorly understood members of the phylum Cnidaria. This member of the Siphonophore order has a morphological colony organization that is very different from others. It contains a gas-filled pneumatophore that allows it to leverage wind and currents, facilitating long-distance dispersal and contributing to its cosmopolitan distribution [1,2,3]. Its long set of stinging tentacles, which can reach up to 30 m in length [4], are densely armed with specialized venom cells called cnidocytes, which discharge and eject a thread that punctures the target and rapidly releases venom into the prey [5]. Consequently, the venom induces paralysis or death in a wide range of vertebrates, including humans, expanding the group of the most venomous animals in the marine world [6,7]. The symptoms observed in humans and vertebrates following envenomation often involve localized effects, such as numbness, paresthesia, and dysesthesias [8]. As the venom spreads across the body, systemic effects can manifest, including neurotoxic symptoms like weakness, nausea, headache, confusion, and drowsiness, along with muscle-related symptoms such as pain, cramps with fasciculations, and respiratory distress [8,9,10].

*Physalis* venom exhibits neurotoxic properties that might directly affect nerve cells or indirectly impact nervous system function through mechanisms such as altering ionic conductance, blocking neuromuscular junctions, or influencing action potentials [7]. The venom disrupts the flow of ions across cell membranes in treated vertebrate cells [11,12], exhibiting a reversible antagonism of nicotinic cholinergic receptors, impacting muscle contractions and nerve signals [13]. Moreover, the venom blocks glutamatergic transmission in neurons and neuromuscular junctions, affecting a crucial excitatory neurotransmitter in the nervous system [14]. The physiological effects of the venom have led to scientific inquiry into its composition. Research has revealed that this venom contains a complex mixture of toxins, including various peptides, proteins, enzymes, and protein inhibitors, along with non-protein compounds (purines, quaternary ammonium compounds, biogenic amines, and betaines) [15,16,17,18]. Even though there have been published studies about *Physalis’*s venom composition, only a few of the proteins have been submitted to public databases (33 protein sequences), mainly mitochondrial. Despite significant research efforts, the specific molecules underlying the potent neurotoxicity of *P. physalis* venom remain understudied.

Cnidaria present great interest in the search for new bioproducts for cosmeceutical and medical applications [19]. The potential applications of venom are further underscored by documented successes in utilizing similar toxins derived from other organisms. Neuroactive peptides targeting ion channels offer promising avenues for developing novel therapeutic strategies against neurological disorders arising from ion channel dysfunction, including neurodegenerative diseases, epilepsy, and chronic pain [20,21]. Several venom-based drugs are already established therapeutic mainstays: “Ziconotide” from cone snails [22], “Bivalirudin” from leeches [23], and “Exenatide” from Gila monster venom [24] have therapeutic applications for severe chronic pain, anticoagulation, and diabetes [25,26]. The presence of potent neurotoxins in *P. physalis* venom constitutes a new avenue for further exploration as a valuable source of novel therapeutic molecules.

The rapid discovery of novel venom peptides requires a cost-effective approach to identify active molecules and characterize their physiological effects. Traditional in vivo testing using mammals is often time-consuming, resource-intensive, and raises ethical concerns [27]. *D. melanogaster* has emerged as a valuable tool due to its several advantages. As a well-understood and cost-effective model organism with a short lifespan, it allows for rapid and high-throughput testing with minimal venom sample requirements [28]. Furthermore, *Drosophila* shares high genetic conservation with humans for key developmental pathways and disease-related genes [29]. This facilitates the extrapolation of findings to human health, while the availability of numerous mutant and transgenic lines enables researchers to dissect the mechanisms of venom toxicity at a deeper level [30]. Beyond genetic manipulability, *Drosophila* offers advantages in its simple, quantifiable behaviors, rapid rearing, and established behavioral assays, further enhancing its suitability for venom research [31]. Moreover, our knowledge of nervous system development and physiology has progressed significantly, largely due to decades of research in genetic, molecular, and behavioral studies performed on simple non-mammalian models. Thus*, D. melanogaster* appeared to be a suitable model for venom toxins screening.

This work focuses on optimizing sensitive endpoints and protocols for toxicological assessments, particularly targeting neurotoxicity and behavioral responses, thereby underscoring the utility of *D. melanogaster* as a model organism. We aim to uncover the potential of *P. physalis* venom as a source of bioactive compounds with promising biotechnological applications.

## 2. Results

### 2.1. Venom Causes Dose- and Time-Dependent Toxicity in the Drosophila Model

Flies exposed to four concentrations of *Physalia* crude venom (Vpp) exhibited a dose- and time-dependent mortality response (Figure 1a). Administration of 240 μg of venom per fly via feeding resulted in 16.67% ± 15.8 mortality after 24 h and 88.89% ± 9.9 mortality at 60 h. Dilutions of Vpp revealed a dose-dependent toxicity effect, with an LD50 of 67.4 μg per fly at the 60 h time point (*p* = 0.05) (Figure 1b). Before dying, treated flies displayed uncoordinated movement and fell when attempting to climb. Notably, this deleterious effect began only after 12 h post-feeding and was consistently observed across all treated flies.

In the treated group exposed to a sublethal dose of 120 µg of venom per fly, disorientation and uncoordinated movement led to climbing difficulties. A significant decrease in the flies’ ability to climb vertically due to negative geotaxis was observed 24 h post-treatment. Treated flies took nearly twice as long to reach the endpoint compared with control flies, with times of 13.3 s ± 6.6 and 7.2 s ± 4.5, respectively (*p* < 0.0001) (Figure 1c). Video recording demonstrate the climbing difficulties faced by treated flies compared with non-treated flies (Appendix A). At 48 h post-treatment, some treated flies were unable to climb higher than 1 cm, repeatedly falling during attempts, while others remained lethargic at the bottom of the tube.

The observed symptoms led us to a deeper investigation into the sublethal effects of the venom toxins.

### 2.2. Venom Induces Movement Alterations

Differences in locomotor activity between treated and non-treated flies were analyzed in real-time using an ethoscopic arena. High-throughput locomotor monitoring greatly facilitated the study of various aspects of locomotor changes. Based on movement recordings, the onset of symptoms such as reduced mobility or paralysis was observed in venom-treated flies (*p* < 0.01) compared with the control group across two time intervals: 0–24 h and 24–48 h. Our findings indicate that symptoms of paralysis or reduced activity began to appear approximately 10 h after venom exposure (Figure 2a). However, when considering total activity over the entire period, the movement was reduced by only 25% in the treated group, likely due to observed hyperexcitability (Figure 2b).

Given that temperature may induce protein denaturation or inactivate certain toxins, we investigated whether heating the venom at 65 °C alters its biological activity (HVpp). Nevertheless, the heat-treated group exhibited a similar pattern of overall movement reduction, with no significant differences between the two groups. No significant differences were also observed in the latency period before the first symptoms appeared among the treated groups. In the time frame between 10 and 24 h, the venom-treated flies exhibited a great movement reduction compared with the activity levels of the control group and displayed symptoms of short periods of hyperactivity followed by paralysis (Figure 2a). These symptoms were transient, with the duration of paralysis ranging from 1 h to more than 5 h. After 48 h, the paralysis symptoms became more permanent, with some flies showing no signs of recovery.

### 2.3. Venom Induces Alterations in Circadian Rhythm

Modifications in circadian rhythm were assessed through continuous monitoring of locomotor activity. To accurately observe circadian timekeeping, two experiments were conducted: in one, the treatment started at the beginning of the light period, and in the other, it began at the end. Control flies exhibited a bimodal pattern of locomotor activity synchronized with 12 h light/12 h dark cycles. No significant differences were noted between the two experiments. During the first 24 h, the treated flies displayed a pattern identical to that of the control group, despite a clear reduction in movement intensity (Figure 3a). After 24 h, a significant decrease in the duration of the active period was observed in the treated groups, with control flies exhibiting locomotor activity for approximately 12 h compared with around 9 h in flies exposed to venom. Regarding food intake, the treated group consumed significantly more than the control group by almost 40% (*p* < 0.05) (Figure 3b). The Vpp-treated group consumed 13.73 µL ± 0.19, while the untreated group consumed 9.5 µL ± 2 per fly. These findings suggest that, even with decreased locomotor activity, the treated flies are compensating by increasing their food consumption, which may have implications for understanding the overall effects of venom on metabolic processes.

### 2.4. Venom Alters the Perception of Temperature and Light

Alterations in the circadian rhythm of treated flies prompted us to investigate the influence of venom on the perception of environmental stimuli. Since light strongly affects the circadian system, a light/dark preference assay was conducted. This assay measured the time each fly spent on the light side of the tube in which it was confined over a period of 1 h (Figure 4a). Venom-treated flies exhibited a greater preference for the light side compared with the control group, spending 60% ± 23% of the time on the light side of the tube, while untreated flies spent only 24% ± 15% (Figure 4b), corresponding to a significant increase of 40% (*p* = 0.0002). This alteration in light perception may indicate changes in sensory neuron responses to various stimuli, which could also affect nociceptive behaviors. To further investigate these sensory alterations, a thermal nociception assay was conducted across different time and temperature conditions (Figure 4c). Tests performed at 44 °C revealed a significant reduction in heat avoidance in treated flies compared with non-treated flies, showing a decrease of approximately 40% (*p* = 0.0009, *n* = 323) (Figure 4d).

## 3. Discussion

To address the growing gap between the increasing discovery of new molecules and the need for in vivo screening and initial characterization, a rapid and cost-effective model for studying toxins is essential. *Drosophila* has been successfully used to evaluate the toxicity of various venoms [32] and for drug screening, offering promising avenues for high-throughput screening of active compounds.

In this study, we investigated the use of *D. melanogaster* to characterize the effects of toxins released by *P. physalis*. To our knowledge, this is the first report testing *P. physalis* venom in the *D. melanogaster* model. *P. physalis* venom induced mortality in a dose- and time-dependent manner when administered orally. At sublethal doses, treated flies exhibited uncoordinated movement and disorientation. A similar phenotype was previously observed in crabs and mice treated with *P. physalis* venom. Crabs injected with *P. physalis* venom made a short, abrupt run, after which they remained motionless [33], consistent with the observed hyperexcitability phenotype that preceded paralysis. These symptoms align with neurotoxic effects such as nausea, headache, and confusion reported in humans stung by *P. physalis* [9,10,11]. Thus, *D. melanogaster* can serve as a valuable model for gaining deeper insights into the neurotoxic effects of *P. physalis* venom.

The paralysis observed in *D. melanogaster* was accompanied by aberrant behavior, likely associated with altered or impaired neural function, suggesting the neurotoxicity of the venom. We hypothesize that changes in movement, circadian rhythm, and heat avoidance may correlate with neuronal activity. Other authors have highlighted the toxic effects of cnidarian venoms, which are attributed to peptides capable of modulating ion channels in the central nervous system [34].

Aligned with this hypothesis, research from the 1990s [13,14] identified toxins in *P. physalis* venom that can affect excitatory neurotransmitters, although the identity of these toxins remains unknown. The P1 toxin exhibits dose-dependent, reversible antagonism of nicotinic cholinergic receptors, impacting muscle contractions and nerve signaling. Toxin P3 (85 kDa) has been shown to block glutamatergic transmission in neurons and neuromuscular junctions, affecting a crucial excitatory neurotransmitter in the nervous system. Additionally, *P. physalis* venom was shown to modulate calcium influx into various types of cells [35]. However, the authors showed that this mechanism does not directly involve (L or T type) calcium channels or ATPase channels [36], and the target responsible for calcium uptake remains unknown. We hypothesize that *P. physalis* venom may contain molecules that interfere with potassium, calcium (other than those mentioned above), or sodium ion channels, resulting in neurotoxicity. It has already been reported that venoms from various cnidarian, as well as venoms from other marine or terrestrial organisms, contain ion channel modulators [34,37,38] that can block nerve conduction, interfering with an electrochemical impulse that propagates the neural pathway [39]. The diversity of ion channels, their wide distribution—including specific expression patterns in the central and peripheral nervous systems—and their vital role in nerve and muscle excitability make them strategic targets for marine toxins [40].

The modulation of ion channels, such as transient receptor potential (TRP) channels, has been implicated in the effects of marine venoms. A reduction in heat avoidance in *Drosophila* treated with a venom peptide from a marine snail was reported by Eriksson [28], suggesting a mechanism involving the modulation of TRP channels responsible for sensing heat stimuli. TRP channels in *Drosophila*, much like nociceptors in mammals, play a role in neurodepolarization under noxious stimuli, transmitting pain-related signals to the central nervous system, potentially influencing pain perception [41].

Moreover, several authors have noted the antinociceptive properties of venoms from wasps [42], snakes [43,44], spiders [45], and cone snails [46,47,48], underscoring the importance of ion channels in mediating the effects of venom. These studies further highlight the strategic value of ion channels as therapeutic targets in venom research and reinforce the importance of investigating *P. physalis* venom in the context of neurotoxic and antinociceptive mechanisms.

According to previous studies, cnidarian venoms are heat labile, which supports the use of hot-water immersion of the sting area in beaten victims [49]. Previous research has shown that treating *P. physalis* venom at 45 °C inhibits its activity, primarily alleviating symptoms [50]. Nevertheless, our findings indicate that the molecules responsible for neuromodulation, as described previously, are not thermolabile at 65 °C. No significant differences in fly behavior and locomotion were observed between flies exposed to naïve or thermally treated venom.

Increased food intake in *Drosophila* was previously noted by Eriksson [28] in flies treated with a venom peptide from a predatory marine snail. An increase in appetite was observed, suggesting that this orexigenic effect is related to modulation via the peripheral nervous system [51], possibly through a neuroendocrinological pathway. In endocrine and neuroendocrine tissues, the excitability of secretory cells is influenced by the repertoire of ion channels that contribute to hormone secretion [52], which may relate to the increased appetite observed in flies treated with *P. physalis* toxins.

Due to the complex mixture of molecules within crude venom tested in this study, the targets and the mechanisms of action of these toxins require further investigation. Nevertheless, this research presents an avenue for discovering novel bioactive molecules from venom with potential pharmaceutical applications as future drugs.

## 4. Conclusions

Although the toxins of *P. physalis* have been recognized for some time, they remain significantly understudied. This study highlights the potential of its venom molecules in revealing their effects on alterations in movement, circadian rhythm, and heat avoidance. These findings underscore the necessity for further investigation into the specific mechanisms of action and targets of these venom toxins. To achieve this, transcriptomic and proteomic studies of *Physalis* venom will be crucial for identifying the molecules underlying the observed phenotypes. Furthermore, this work establishes key endpoints and demonstrates the suitability of *D. melanogaster* as a model organism for analyzing the effects of *P. physalis* venom. In addition, this study contributes to a deeper understanding of the activity of *P. physalis* crude venom, paving the way for future investigation into novel bioactive molecules, particularly neuroactive compounds, encouraging further exploration of therapeutic compounds. Therefore, this research could significantly stimulate interest in the therapeutic potential of marine biodiversity.

## 5. Materials and Methods

### 5.1. Collecting and Fixing

*Physalia* specimens were collected from locations along the Azorean coast of São Miguel Island, situated in the middle of the Atlantic Ocean. Information on when to collect large quantities of *P. physalis* was obtained from the official site of the Portuguese Institute of the Sea and the Atmosphere [53], which is dedicated to sightings of jellyfish species. Juvenile and adult specimens were collected alive and fresh from the surf, separated by tentacles and pneumatophore, and transferred directly to the lab in sterile tanks with seawater at 16–18 °C for examination. Afterward, the material was washed with distilled water, frozen, and stored at −80 °C.

### 5.2. Biological Materials

*D. melanogaster* (strain W118) was reared at 25°C on standard food (agar medium containing sucrose). All experiments were performed on 5-day-old mated females. The conditions for all insects were 12/12 h light/dark cycles and 50% humidity.

### 5.3. Venom Extraction

Venom extraction was performed using glass beads using a method described by Toubarro (2023) [54]. In brief, 1 cm of tentacle samples was incubated with 20 mM PB at pH 7.4 and shaken in a mini-bead mill for 1 min, repeated five times with intermittent cooling on ice. The homogenate was then transferred to a new tube and centrifuged at 10,000× *g* for 5 min, and the supernatant was used. Quantification of proteins in the total venom sample was performed using a Nanodrop.

### 5.4. Dose-Dependent Response

To determine the LD50, 1- to 5-day-old females of *D. melanogaster* were used. After anesthetizing the flies by cooling them on ice for 10 min, they were separated into 15 tubes (10 flies in each). A sucrose solution at 0.1 M was supplemented with varying venom doses (300 µg, 600 µg, 1200 µg, and 2400 µg) to feed the flies. Four capillaries were provided in each tube to deliver the treatment to the flies. Three technical replicates of each tube were performed. The tubes were sealed with parafilm, and the assays were performed at 25 °C, with a 12 h light/dark cycle and 50% humidity, for 60 h. The LD50 was estimated using a nonlinear regression model in GraphPad Prism 8.0.1.

### 5.5. Negative Geotaxis Assay

The locomotor capacity was evaluated using the negative geotaxis assay, as described previously [55,56], with some minor modifications. Ten adult female flies (5 days old) were immobilized on ice and placed separately in a polypropylene tube (length: 25 cm; diameter: 1.5 cm). A sucrose solution at 0.1 M was supplemented with 1200 µg of venom to feed the flies, complemented with blue dye. For the control group, PBS (20 mM, pH 7) was used instead of venom. Four capillaries were provided in each tube to deliver the treatment to the flies. A total of 80 flies were tested, 10 per tube, and treated for a 24 h period. Afterward, the flies were gently tapped to the bottom of the tube, and the time for each fly to reach the marked point at the top was recorded, obtained from three independent experiments. Results are presented as the time each fly needed to reach the top. Flies that were unable to reach the top were considered to have a maximum measured time of 20 s.

### 5.6. Real-Time Locomotion Assay

To study the *D. melanogaster* circadian rhythm, locomotor capacity and behavior were measured in real-time using an ethoscope, as described previously [57]. Flies were recorded using the Official Raspberry Pi 8MP NOIR V2 Camera Module to capture and process infrared-illuminated video up to a resolution of 1920 × 1080 pixels at 30 frames per second [57].

For the assays, 5-day-old female flies were used, anesthetized by cooling on ice, and then placed in individual tubes within an ethoscopic “sleep arena”. Each tube was sealed with a cotton plug at one end and parafilm at the other end, outside the arena. Each fly was fed by one capillary with sucrose solution at 0.1 M, supplemented with 120 µg of venom, complemented with blue dye. For the control, flies were fed with PBS 20 mM, pH 7 instead of venom. Venom treated for 15 min at 65 °C (HVpp) was also tested to investigate whether heating alters its biological activity. The fresh solution was replaced every day. The experiment was performed under controlled conditions, at 25 °C, 12/12 h light/dark cycles, and 50% humidity. A total of 4 biological replicates were performed using 80 flies. All data obtained from the ethoscope recordings were analyzed using R 4.2.2 (2024.04.2+764) software with the Rethomics package. The data were pre-processed using the sleep-annotation function of the sleep R package, and activity data were analyzed in 10 s windows. Sleep was defined as a state in which the fly was inactive for more than five minutes.

### 5.7. Food Consumption

During the study of the venom’s influence on *D. melanogaster* locomotor activity, we compared food intake among the different groups. While recording each fly in the ethological arena, we measured food intake using glass microcapillary tubes at 24 and 48 h time points. Three variants were compared: control (C), venom (Vpp), and heated venom sample (HVpp). A total of approximately 60 flies were tested in 3 biological replicates.

### 5.8. Nociception Study

To investigate thermal nociception, we used the assay proposed by Neely and Keene (2011) [58] with minor modifications. One- to five-day-old female flies were placed into a behavioral chamber (35 mm × 10 mm Petri dish) and sealed with parafilm. The flies were allowed to rest for a predetermined amount of time. The chambers were then floated in a water bath at 40–45 °C for 1–5 min. Assays were performed in the dark. Afterward, the chambers were removed from the water bath, and the number of “incapacitated” (immobilized by the temperature) and naïve flies was recorded. The fraction of avoidance was calculated as described by Neely and Keene (2011) [58], following the equation: (total flies − immobilized flies)/(total flies). A total of approximately 300 flies were tested, with 10 flies in each chamber across 6 biological replicates.

### 5.9. Light–Dark Preference Assay

For the light attraction behavior assay, adult female flies (5 days old) were immobilized on ice, placed in a polypropylene tube, and fed with sucrose solution at 0.1 M supplemented with 1200 µg of venom, as described previously. Six hours after feeding, the flies were placed into plastic tubes plugged with cotton to be real-time recorded in an ethoscopic arena as described before, with slight modifications. Half of each tube was covered with aluminum foil to create a dark space. The recorded movement was treated as an indicator of light preference. Fly behavior was recorded for 1 h using the same video recording setup as described above. A total of 60 flies across 3 biological replicates were analyzed.

### 5.10. Data Analysis and Statistic

Collected data were analyzed using R 4.2.2 (2024.04.2+764), GraphPad Prism 8.0.1, and Microsoft^®^ Excel (Version 2410 Build 16.0.18129.20158). Differences were considered statistically significant at *p* < 0.05. Outliers and dead flies during the initial period were excluded. Locomotor activity traces and plots were generated in R Studio using the ggplot2 package. Statistical analysis consisted of one-way ANOVA for significance, and Tukey’s multiple comparison test was performed on the aggregated data from the ethological analysis. For other assays, unpaired *t*-tests and ANOVA were performed using GraphPad Prism. Unless otherwise indicated, data are represented as mean values ± SD. In this work, an AI-based tool, ChatGPT (GPT-4, October 2024 version) from OpenAI, was used to check grammar accuracy and improve the clarity and structure of the text.

## Figures and Tables

**Figure 1 toxins-16-00491-f001:**
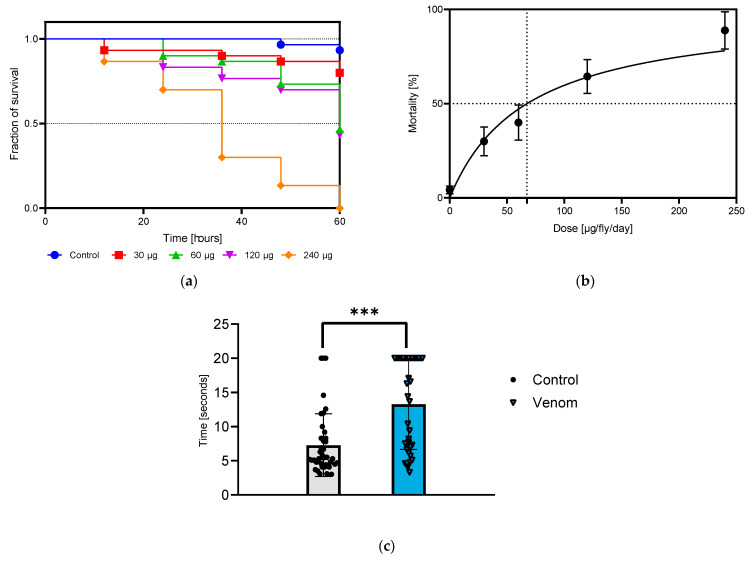
(**a**) Survival of flies treated with different doses of venom. (**b**) Dose-dependent responses at 60 h. Doses are presented in μg/fly/day, and values correspond to the mean ± SEM. (**c**) Negative geotaxis assay comparing venom-treated flies to control flies. *** Highly significant difference (*p* < 0.001, Student’s *t*-test).

**Figure 2 toxins-16-00491-f002:**
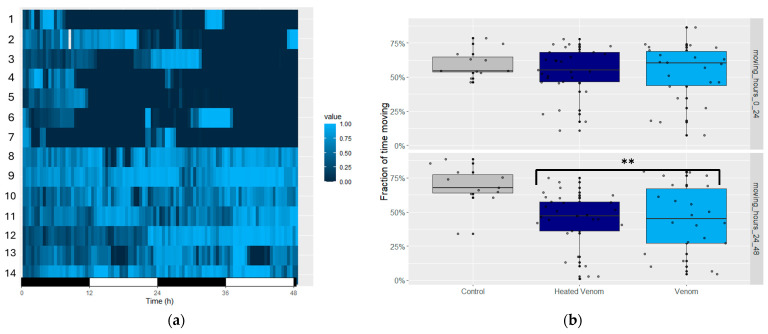
(**a**) Heat map representing the detailed activity of each fly over 48 h, comparing treated and non-treated flies. The x-axis represents time in hours (0 to 48 h), and the y-axis lists the experimental replicates, where 1–7 are venom-treated flies, and 8–14 are controls. The color gradient indicates the level of activity, with darker shades representing less movement and lighter shades (blue) indicating higher activity levels. (**b**) Box plot representing the fraction of time the *Drosophila* spent moving for each group, *n* = 84. ** Indicate highly significant difference (*p* < 0.01, ANOVA).

**Figure 3 toxins-16-00491-f003:**
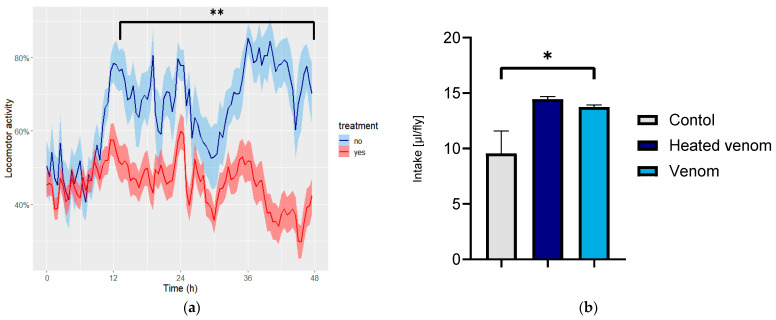
(**a**) Percentage of locomotor activity (*Y*-axis) over time (hours, *X*-axis), with the blue line indicating non-treated flies and the red line representing treated flies. Shaded areas denote standard error. (**b**) Food intake between treated and non-treated flies, represented in µL per fly. * Indicate significant differences (*p* < 0.05) and ** highly significant differences (*p* < 0.01), as determined by ANOVA.

**Figure 4 toxins-16-00491-f004:**
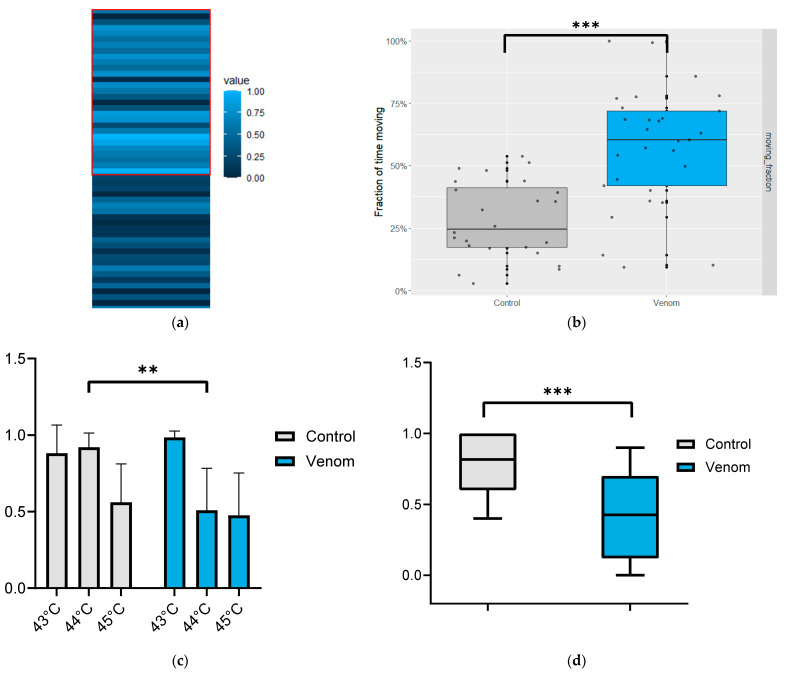
(**a**) The heat map represents the activity of each fly over a 1 h period. Each bar represents the time each fly spent on the light side of the tube. Individuals treated with venom are enclosed in a red frame. (**b**) Box plot represents the light-side preference for treated and non-treated flies. (**c**) Heat avoidance across various temperatures during 3 min incubations. The *Y*-axis represents the relative number of incapacitated flies compared with the control, with 1.0 corresponding to 100 percent. (**d**) Comparison of heat avoidance behavior at 44 °C between treated and non-treated flies over 3 min (*n* = 323). The *Y*-axis represents the relative level as a fraction of control, where 1.0 is equal to 100 percent. ** Highly significant difference *p* < 0.01, *** highly significant difference *p* < 0.001 (Student’s *t*-test, ANOVA).

## Data Availability

The raw data supporting the conclusions of this article are available on request from the corresponding author due to the inclusion of raw data that has not been published in this article and is planned for release in a forthcoming publication.

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
