# Peer review of "Revealing the Bioactivities of *Physalia physalis* Venom Using *Drosophila* as a Model"

_toxins, 2024, doi:10.3390/toxins16110491_

Round 1

Reviewer 1 Report

Comments and Suggestions for Authors

Revealing the Bioactivities of Physalia physalis venom using Drosophila as a Model

I find this manuscript trying to address the relevant issue and the paper to be generally scope of the journal. The manuscript is well-structured and presents valuable findings in the field of Bioactivities of Physalia physalis venom. However, it needs more significant improvement. I have number comments and suggestions for improvement of the paper and have provided here with minor revisions.

1.     Abstract need more clear and concise format.

2.     The keywords should be more relevant.

3.     What is the novelty of this work.

4.     Figure 2(a) is not clear

5.     Method and methodology should be improved with better explanation.

6.     Justify the Quality assurance and quality control in the Methods and methodology section.

7.     There are some very long paragraphs that are hard to read, these could benefit from being split up into separate paragraphs based on topic, with topic sentences to introduce each one.

Conclusions is not just about summarizing the key results of the study, it should highlights the insights and the applicability of your findings/results for further work. The author's own opinions too little summary about future prospects points of research in the manuscript

Author Response

  1. Abstract need more clear and concise format.

We acknowledge the reviewer’s feedback and have revised the abstract to enhance its clarity.

  1. The keywords should be more relevant.

We acknowledge the reviewer’s suggestion adding relevant keywords.

  1. What is the novelty of this work.

We appreciate the reviewer’s feedback and have highlighted the relevance and novelty of the work at the end of the introduction by adding a paragraph (lines 99 to 103). We have also highlighted the relevance and novelty of the work in the conclusion section, specifically within lines 284 to 294.

  1. Figure 2(a) is not clear

Figure 2 and its legend have been revised as suggested, with enhancements to clearly highlight the treatment and control groups.

  1. Method and methodology should be improved with better explanation.

The improvements have been added to the methods in sections Dose-dependent response), Negative geotaxis assay, as well as in the Real-time locomotion assay, Nociception study, and Light-dark preference assay.

  1. Justify the Quality assurance and quality control in the Methods and methodology section.

We agree with the suggestion and have added information in the materials and methods section regarding the controlled conditions during experiments, including the use of four biological replicates and the total number of flies used.

  1. There are some very long paragraphs that are hard to read, these could benefit from being split up into separate paragraphs based on topic, with topic sentences to introduce each one.

We appreciate the reviewer observation and have revised parts of the manuscript to improve readability and clarity.

  1. Conclusions is not just about summarizing the key results of the study, it should highlights the insights and the applicability of your findings/results for further work. The author's own opinions too little summary about future prospects points of research in the manuscript

We appreciate the reviewer suggestion and have addressed this by adding a section (lines 286–292) in the conclusion. This addition highlights the relevance and potential applicability of our findings for future research, providing insights into how this work can guide further studies and advancements in the field.

Reviewer 2 Report

Comments and Suggestions for Authors

Authors investigated the bioactivities of Physalia physalis venom using Drosophila melanogaster as a model organism. They demonstrated dose- and time-dependent mortality in flies exposed to the venom, along with behavioural changes such as uncoordinated movement, disorientation, and alterations in circadian rhythm. The venom also affected light preference and heat avoidance behaviours. The authors suggest that these effects indicate neurotoxicity, potentially due to altered neuronal function. Overall, the study highlights the potential of D. melanogaster as an efficient model for venom activity screening and contributes to the understanding of P. physalis venom. Minor comments,

1. Were any molecular or cellular assays performed to investigate the mechanisms underlying the observed behavioural changes?

2. What are the potential limitations of using Drosophila as a model for studying the venom effects?

3. Were any attempts made to fractionate the venom to identify specific components responsible for the observed effects?

4. Were any sex-specific differences observed in the response to venom treatment?

5. How do you plan to address the potential ethical concerns of using venom-based compounds for therapeutic applications?

6. What follow-up studies would you propose to further elucidate the mechanisms of action of P. physalis venom?

7. How do the observed effects in Drosophila compare to those reported in vertebrate models?

8. Were any controls performed to rule out potential effects of the venom extraction process on the observed toxicity?

Author Response

  1. Were any molecular or cellular assays performed to investigate the mechanisms underlying the observed behavioural changes?

In this study, we focused on phenotyping assays to characterize the observed behavioral changes. We agree with the reviewer’s suggestion, and in future work, we aim to investigate the molecular mechanisms underlying these phenotypes, building on the results we have obtained here.

  1. What are the potential limitations of using Drosophila as a model for studying the venom effects?

Drosophila provides valuable insights into neurotoxicity, with several observed phenotypes aligning with symptoms seen in vertebrates, indicating its utility as a model for preliminary assessments of venom effects. However, we acknowledge the limitation that toxin specificity may be more pronounced in vertebrates, given their closer evolutionary relationship with typical vertebrate prey of P. physalis. Nevertheless Drosophila serves as an important and accessible model for uncovering general toxic effects and neurotoxic phenotypes, providing a valuable platform for initial insights that can guide further research into the specific mechanisms of venom action.

  1. Were any attempts made to fractionate the venom to identify specific components responsible for the observed effects?

This is a preliminary study aimed at highlighting the neurotoxic and nociceptive effects of the venom, providing a foundational understanding that opens doors for future fractionation and purification studies targeting specific venom components responsible for these effects.

  1. Were any sex-specific differences observed in the response to venom treatment?

Due to technical issues, such as males entering the food-supplied capillary because of their smaller size, we decided to proceed with only females in subsequent assays. Nonetheless, no significant differences were observed between males and females in the initial tests (data not shown).

  1. How do you plan to address the potential ethical concerns of using venom-based compounds for therapeutic applications?

We understand the reviewer's concerns. In lines 75-81, we have provided examples of drugs derived from venom toxins that are currently in clinical or commercial phases, illustrating how venom-based compounds can be developed responsibly and ethically for therapeutic applications.

  1. What follow-up studies would you propose to further elucidate the mechanisms of action of P. physalis venom?

In lines 286-292 of the conclusion, we have added information about ongoing studies aimed at further elucidating the mechanisms of action of P. physalis venom. These follow-up studies focus on identifying specific toxins within the venom to better understand their roles and effects.

  1. How do the observed effects in Drosophila compare to those reported in vertebrate models?

We agree that this is a crucial topic. In the discussion section (lines 221 to 227), we compare the phenotypes observed in Drosophila with those reported in vertebrate models, emphasizing the similarities in neurotoxic effects.

  1. Were any controls performed to rule out potential effects of the venom extraction process on the observed toxicity?

This venom extraction method is widely used by other researchers (doi.org/10.1186/s40409-017-0125-8; DOI:10.1016/j.toxicon.2004.04.008; doi: 10.3390/md14040075), and we selected it specifically for this assay. However, in our previous work (https://doi.org/10.3390/IECT2023-14810), we tested different extraction methods and compared the protein profiles, which revealed variations. This previous analysis helped ensure that our chosen extraction method was appropriate for studying the observed toxicity.

Reviewer 3 Report

Comments and Suggestions for Authors

This manuscript utilizes Drosophila as a model to evaluate the bioactivities of Physalia physalis. Some issues need to be addressed as follows.

1. A positive control, for example, a toxin that is known to show bioactivities in Drosophila, needs to be included in the experiment and discussed in the manuscript as a comparison.

2. The manuscript needs to be carefully proofread especially the fonts need to be the same for publication.

3. In Figure 4c and 4d, titles for the Y axis should be added.

Author Response

  1. A positive control, for example, a toxin that is known to show bioactivities in Drosophila, needs to be included in the experiment and discussed in the manuscript as a comparison.

We understand the reviewer’s concern. In this preliminary study, our focus was to explore the overall activity of the venom, though the specific toxins involved are not yet identified. Therefore, it was not possible to include a control toxin with known bioactivity in Drosophila for direct comparison.

  1. The manuscript needs to be carefully proofread especially the fonts need to be the same for publication.

The fonts have been standardized throughout the manuscript, and both the abstract and conclusion have been carefully proofread for clarity and consistency.

  1. In Figure 4c and 4d, titles for the Y axis should be added.

We agree with the reviewer and have clarified the figure legends. The Y-axis represents the relative number of incapacitated flies compared to the control, with 1.0 corresponding to 100 percent.